# DNA and RNA Binding Proteins: From Motifs to Roles in Cancer

**DOI:** 10.3390/ijms23169329

**Published:** 2022-08-18

**Authors:** Ondrej Bonczek, Lixiao Wang, Sivakumar Vadivel Gnanasundram, Sa Chen, Lucia Haronikova, Filip Zavadil-Kokas, Borivoj Vojtesek

**Affiliations:** 1Research Centre for Applied Molecular Oncology (RECAMO), Masaryk Memorial Cancer Institute (MMCI), Zluty Kopec 7, 656 53 Brno, Czech Republic; 2Department of Medical Biosciences, Umea University, 90187 Umea, Sweden

**Keywords:** DNA/RNA binding protein, mutation, cancer, targeted treatment, biomarkers

## Abstract

DNA and RNA binding proteins (DRBPs) are a broad class of molecules that regulate numerous cellular processes across all living organisms, creating intricate dynamic multilevel networks to control nucleotide metabolism and gene expression. These interactions are highly regulated, and dysregulation contributes to the development of a variety of diseases, including cancer. An increasing number of proteins with DNA and/or RNA binding activities have been identified in recent years, and it is important to understand how their activities are related to the molecular mechanisms of cancer. In addition, many of these proteins have overlapping functions, and it is therefore essential to analyze not only the loss of function of individual factors, but also to group abnormalities into specific types of activities in regard to particular cancer types. In this review, we summarize the classes of DNA-binding, RNA-binding, and DRBPs, drawing particular attention to the similarities and differences between these protein classes. We also perform a cross-search analysis of relevant protein databases, together with our own pipeline, to identify DRBPs involved in cancer. We discuss the most common DRBPs and how they are related to specific cancers, reviewing their biochemical, molecular biological, and cellular properties to highlight their functions and potential as targets for treatment.

## 1. Introduction

From DNA to RNA (and vice versa) and from mRNA to protein is the central dogma of how genetic information is stored, processed, and transmitted in biological systems, but these three molecular entities also regulate each other. Understanding the regulatory mechanisms between nucleic acids and proteins offers pathways for therapeutic intervention. In recent years, attention has focused on DNA and RNA binding proteins (DBPs and RBPs) and especially on proteins that have dual properties, namely DNA and RNA binding proteins (DRBPs) [1,2,3]. The dysregulation of these proteins is found in various human diseases, most notably in cancer. The purpose of this review is to provide a comprehensive view of DRBP structures and properties, their roles in cellular signaling pathways, and the relationships between the disruption of their biological activity and the onset of disease, with an emphasis on cancer.

### 1.1. DNA Binding Protein Motifs

The most extensive and well-known group of nucleic acid-binding proteins are DBPs, especially transcription factors (TFs). Binding to DNA occurs via two basic principles: (i) direct or water-mediated hydrogen bonds and/or van der Waals interactions (mainly in the major groove of B-DNA) and (ii) contact with sequence-specific DNA motifs [4,5]. Based on these two principles, together with protein–protein interaction motifs, DBPs can be divided into well-established families, including helix-turn-helix (HTH), leucine zipper (ZIP), basic helix-loop-helix (bHLH), proteins with β-sheets, the homeodomain complexes (HOX), and the superfamily of high mobility group (HMG) proteins. Each of these families is characterized by a high similarity in the primary functional (active) domain of the proteins [6].

HTH proteins are an ancient family characterized by two α-helices separated by a linker (turn). This feature is evolutionarily conserved across prokaryotes and eukaryotes and further enriched by additional elements and/or amino acid sequences (α-helices, β-sheets, and β-barrels) [7,8]. The basic motif is part of a DNA-binding domain containing a tri-helical structure with a typical sharp turn between the 2nd and 3rd helices. The 3rd helix is responsible for the interface between the protein and the major groove of B-DNA. Secondary stabilizing contacts with DNA may be mediated by residues outside the HTH active core [9,10,11,12].

Like HTH, bHLH proteins have a common ancestor consisting of the DNA-binding domain alone, which later incorporated additional flanking sequences [13]. The common ancestors of these two families differ in sequence signatures [8]. Generally, bHLH proteins are transcriptional regulators unique to eukaryotic organisms and are involved in developmental pathways. Several classifications have been proposed. Atchley and Fitch [14] defined five groups (A-D and unclassified) based on phylogenetics and binding to DNA at the hexanucleotide E-box (CANNTG). More recently, Ledent et al. [15] suggested seven groups (A–F and Atonal). Two structural components can be distinguished, in which the basic residues allow binding to DNA, and the hydrophobic HLH domain mediates protein interactions and homo- or heterodimer formation. Dimerization creates two amphipathic α-helices separated by a variable loop [16,17]. A ZIP may be present outside the HLH domain as an additional dimerization and interaction motif [18,19]. Beside α-helices (in HTH and bHLH), less common motifs are two stranded anti-parallel or three-stranded β-sheets that come into contact with the major groove of DNA [20,21].

Historically, an adjacent basic region and 4 to 5 recurring leucines every 7 residues has been considered as the primary motif of the ZIP family of proteins. Precisely spaced leucines allow for the formation of stable dimers of α-helices. The structure can be imagined as a Y-shaped “scissor grip,” which is where the name of the motif comes from [18,22,23]. A ZIP motif occurs only rarely as a solitary motif, and is more often found as a cooperating member of supramolecular complexes. A ZIP is present in proteins that also contain zinc fingers or homeodomains, supporting the idea that the motif act as an independent dimerization region for different types of DNA-binding domains [7].

Members of the HMG superfamily are involved in eukaryotic DNA transcription, replication, recombination, and repair. Proteins in this superfamily typically recognize non-B-DNA conformations. The HMG superfamily consists of three families, each of which contain unique motifs, but all specifically affect chromatin fibers [24]. The positively charged AT hook motif (HMGA), consisting of 9 amino acids, tethers proteins to (and unbends) the minor groove of DNA. The second family, HMGB, is characterized by a sequence of approximately 75 amino acids and has a characteristic, twisted, L-shaped fold formed by three α-helical regions. HMGB proteins also bind to the minor groove of DNA, but have the opposite effect, causing DNA bending. The third family, HMGN, has a nucleosomal binding domain, originally found in two archetypal HMG proteins: HMG-14 and HMG-17. These two proteins are present in the nuclei of all higher eukaryotes and recognize the 146-bp nucleosome core (chromatin fiber). They cooperate to bind nucleosomes, always two molecules of HMG-14 or HMG-17 [25,26,27,28].

### 1.2. RNA Binding Protein Motifs

RNA binding motifs are diverse and although a large number have been discovered, they cover only about a quarter of all RBPs, suggesting that other motifs are yet to be revealed [29].

Although the properties of known RBP motifs are similar to those of DBPs, there are some specific differences. One obvious example is the presence of 2′-OH groups in RNA, which is used for binding by approximately 20% of all RBPs [30]. In addition, the base-pairing moieties of RNA are more commonly used for protein contacts than are those in base-paired DNA. In this respect, RBPs need to accommodate the complex structures of RNAs (tertiary conformations), whereas the DNA helix is often surrounded by DBPs [31].

Besides hydrogen bonds and van der Waals forces, π interactions (π stacking) are involved in protein-RNA interactions [32]. π-interactions provide non-covalent contacts between a nitrogenous base ring and a π-containing amino acid (with aromatic or charged residues) and hydrophobic contacts (between RNA bases and hydrophobic side chains of amino acids). π-interactions are also abundant in DNA, but their impact on protein binding is not fully understood [33].

RNA-recognition motifs (RRMs) orchestrate most post-transcriptional gene processes (mRNA and rRNA processing, RNA export and stability, as well as translation). RRMs are typically composed of approximately 90 amino acids forming a four-stranded β-sheet packed against two α-helices, which interact with single-stranded RNA (ssRNA) through sequential stacking interactions and hydrogen bonds with 2 to 8 nucleotides [34,35].

Proteins that specifically bind to double-stranded RNA (dsRNA) regions (e.g., in miRNA or siRNA processing) contain an eponymous domain (double-stranded RNA-binding domain; dsRBD), folded from antiparallel β-sheets flanked by α-helices on one face (α1β1β2β3α2). It was thought that dsRNA recognition is based on the structural conformation of the A-RNA helix, rather than sequence specific recognition. However, it is now apparent that sequence-specific contacts also play a role [35,36]. dsRBDs exclusively bind to dsRNA rather than dsDNA because of the chemical properties of RNA (presence of 2߰-OH) and the three duplex grooves of A-form dsRNA. However, dsRBDs can also affect RNA-DNA duplexes [37].

Examples of typical sequence-specific RBPs are PUF domain-containing proteins (or Pumilio homology domain). This large domain has eight α-helices that interact with one nucleotide per one helical repeat through hydrogen bonds and stabilization by stacking, distinguishing up to eight nucleotides on ssRNA [38,39]. Artificial interventions through protein engineering have resulted in modification of the motif and the selection of “tailor-made” binding sequences. The ability to produce such manipulations has potential uses in biotechnology and medicine [40].

### 1.3. DNA/RNA Binding Protein Domains

Despite the obvious differences between DNA and RNA, approximately 2% of human proteins are able to interact with both DNA and RNA [3]. This dual activity is either because they contain a motif that is intrinsically able to bind both, or because they contain separate motifs for each function.

The dynamic, reciprocal flow of interactions can then be divided into three levels:(i)Regulation of DNA/(m)RNA function/fate by binding of DNA/(m)RNA to protein.(ii)Regulation of protein function/fate by binding to DNA/(m)RNA substrates.(iii)Competition of binding to DNA/(m)RNA with other binding partners, thus controlling a particular set of DNAs/(m)RNAs in response to the level of other molecules [3,29,41].

A typical example of dual functional domains is the cold shock domain (CSD), consisting of five antiparallel β-strands that form a β-barrel. The CSD binds with relatively high affinity to ssRNA and ssDNA and typically contains RNP (ribonucleoprotein) 1 and 2 RNA-binding motifs [42]. The overall positive charge typical of the CSD provides non-specific electrostatic interaction with negatively charged nucleic acids, while aromatic amino acid side chains stabilize this binding through hydrophobic and stacking interactions [43]. CSD proteins (CSPs) are found in eukaryotes and prokaryotes. CSPs contain additional domains (zinc finger, glycine-rich, or alanine/proline domains) to provide a large group of multifunctional proteins. A typical representative eukaryotic CSP is Y-box binding protein 1 (YB-1), which is well known for its function in oncogenesis [44,45].

The discovery of the linear polar recognizers, zinc fingers, was preceded by biochemical studies on the 5S RNA genes of *Xenopus laevis*, which are transcribed by RNA polymerase III. Transcription factor IIIa (TFIIIA) is required to initiate this process. Miller et al. [46] noticed a 30-amino-acid repeat in TFIIIA, which subsequently resulted in a general pattern for this motif—a tetrahedrally coordinated single atom of zinc between two invariant pairs of cysteines and histidines (C2H2), which forms a typical “finger.” The zinc finger binding machinery (domain) usually has a modular character, with many zinc finger units connected by linkers, and is present in RBPs, DBPs, and DRBPs [47,48]. Interestingly, zinc may be coordinated with two pairs of cysteines, and the whole motif is not always directly involved in binding to the nucleic acid, but may be a necessary structural modifier to allow the interaction [7]. A zinc finger may also be part of a homeodomain complex, such as a steroid receptor, where the zinc is coordinated by four cysteines followed by an extended α-helical region [49].

The K homology (KH) domain was identified in heterogeneous nuclear ribonucleoprotein K (HNRNPK) by Siomi et al. [50]. Its single-stranded sequence-specific domain binds nucleic acids with a conserved “GxxG” loop [51]. The KH domain is composed of three α-helices packed onto the surface of a central antiparallel β-sheet. Eukaryotic type I and prokaryotic type II KH domains share a βααβ core structure, with α and β regions on the C- or N-terminus, respectively. The recognition of four nucleotides by a groove in the protein is mediated by hydrophobic and hydrogen bonds. The KH domain is present in many proteins and typically recognizes ssRNA or ssDNA [52,53,54].

## 2. Cancer-Associated DNA/RNA Binding Proteins

To generate a meaningful list of DRBPs involved in cancer, we performed searches in four databases—UniProt [55], Gene Ontology (GO) [56], the Eukaryotic Nucleic Acid Binding Protein Database (ENPD) [57], and the RNA-Binding Protein Database (RBPDB) [58]—containing records of DNA and/or RNA binding properties and with overlapping or distinct annotations. Data were compared using Venn diagrams based on the names and synonyms of the proteins. Surprisingly, only six proteins were identified in all four of these databases (Figure 1). The underlying reason for this low number is that although the databases should be based on the same global data, there are discrepancies due to different classification methods, time of database update, and different levels of evaluation for experimental evidence. We should note that “experimental evidence” for DNA/RNA binding in databases is not always proof of direct binding, and DRBPs are often classified on the basis of participation with other DRBPs, where they act as part of a multi-protein complex, or by high throughput methods without further investigation.

For these reasons, this stringent approach produced an unrealistically low number of DRBPs; therefore, we created our own search pipeline to find intersections between annotated DRBPs and those with roles in cancer. We searched the GO database using the QuickGO browser [59,60] including these criteria—Taxon: 9606 Homo sapiens; Proteins: Reviewed (Swiss-Prot) and Reference Proteomes (Gene centric, canonical); GO terms: DNA binding (GO:0003677) and RNA binding (GO:0003723); Aspect: Molecular function; Evidence: Experimental evidence used in manual assertion (ECO:0000269). Proteins meeting these criteria were then searched for mutation in The Cancer Genome Atlas (TCGA) PanCancer Atlas (cBioPortal) [61,62], and we selected proteins altered in more than 3% of all cancers. This set of proteins was then analyzed using STRING 11.0b [63], with a minimal interaction score of 0.700 and excluding disconnected nodes. The final plot of 21 interacting proteins with roles in cancer is shown in Figure 2.

More detailed analyses of individual protein clusters within the network showed specific patterns of cancer-associated alterations. For example, analysis of a tight 5-protein network (HNRNPL, HNRPU, ILF2, ILF3, DDHX9) showed that gene copy number variations (CNV) dominate in ovarian serous cystadenocarcinoma, whereas mutations are the most frequent aberration to these genes in uterine corpus endometrial carcinoma (Figure 3).

On the other hand, proteins such as p53 are considered as an interaction hub and are frequently mutated across most cancer types. Besides mutations as the main factors causing dysfunction (gain or loss of activity), SOX2 shows that its CNV might play a key role to affect DNA/RNA binding. Therefore, there is evidence that it can be tissue-dependent, since it shows different CNV in different cancers (see Figure 4). These results indicate that DRBPs in cancers affect the regulation of cellular processes by different mechanisms. 

Finally, we investigated whether these genes show specific alteration types across cancer. Figure 5 shows that each of the 21 proteins are affected by specific types of genetic alteration. Taken together, these data confirm that alterations to DRBPs are non-random, implying that a distinct type of alteration to each protein (gain or loss of specific activity) is involved in the oncogenesis of particular types of cancer.

In the following paragraphs, we group the 21 DRBPs according to their main known function and we review their biochemical, molecular biological, and cellular functions, as well as their potential to act as targets for therapeutic intervention. An overview and a list of details are provided in Appendix A.

### 2.1. Molecular Motors

The two largest families of DExD (DDX)- and DExH (DHX)-box RNA helicases belong to the helicase superfamily 2 and are named after their Asp-Glu-X-Asp/His motifs. DDX and DHX members play crucial roles in RNA metabolism across all eukaryotic cells. DDX and DHX proteins contain two recombinase A-like (RecA) domains, on which nucleoside triphosphate (NTP) binding/hydrolysis and nucleic acid binding sites lie. N-and C-terminal extensions promote specific interactions of each member [64]. DDX3X is ubiquitously expressed in all *Eutheria* and is implicated in IFN-α and IFN-β induction upon virus infection [65], in Wnt/β-catenin signaling [66], and in RNA interference (RNAi) [67]. Unbalanced DDX3X expression occurs in many cancer types, and DDX3 usually, but not always, provides oncogenic effects, suggesting it as a target for inhibition with small molecules [68,69].

The first known function of DDX60 was in the RIG-I-like receptor-mediated pathways that induce inflammatory cytokines after viral infection [70]. DDX60 has not been studied in detail in cancer, although lower levels in breast cancer are associated with radiosensitivity [71], and high a level of DDX60 is a proposed biomarker and prognostic factor for oral squamous cell carcinoma [72], as well as an indicator of response to immune checkpoint inhibitors in glioma [73].

DHX9 and DHX36 are G-quadruplex- (G4) and polysome-associated proteins [74,75]. DHX9 maintains translation of α1 and α2 type I collagen, which is the most abundant protein in humans [76]. Its association with translational control protein 80 (TCP80) following DNA damage and its co-overexpression leads to enhanced expression of *TP53* via regulation of its internal ribosome entry site [77]. DHX36 plays a role in mediating antiviral innate immunity as a sensor for viral nucleic acids [78], recognizes non-methylated DNA together with DHX9, and can activate antimicrobial responses [79]. DHX36 is also associated with the translocation of miRNAs and interactions with Argonaute (AGO) proteins [80] and may regulate TP53 pre-mRNA 3′-end processing after DNA damage [81]. DHX9 and DHX36 are upregulated in advanced stage colorectal cancer (CRC) [82]. DHX9 promotes migration of lung adenocarcinoma cells [83], whereas *DHX36* knockdown increases migration and proliferation and reduces chemotherapeutic responses in non-small-cell lung carcinoma (NSCLC) [84]. *DHX36* is commonly overexpressed in head and neck cancers [85].

Chromodomain Helicase DNA Binding Protein 3 (CHD3, also called Mi-2α) belongs to the family of ATP-dependent chromatin remodeling proteins and forms part of the nucleosome remodeling and histone deacetylase (NuRD/Mi-2) complex [86]. The RNA binding properties of CHD3 were identified by ChIP-seq [87]. Inherited missense mutations in the helicase domain of *CHD3* affect chromatin remodeling activity and are associated with neurodevelopmental disorders [88]. Many CHD proteins act as context-dependent tumor suppressors and although not as common as other members, somatic mutations in *CHD3* are reported to be associated with prostate cancer, breast cancer, gastric cancer, and CRC [89,90].

UPF1 (regulator of nonsense transcripts 1) is a highly processive helicase, using its two RecA-like domains for RNA and DNA unwinding in a 5′->3′ manner [91,92]. UPF1 contains a cysteine-histidine-rich (CH) domain at its N-terminus that reduces its helicase activity, and this inhibitory effect is released by interaction with proteins such as UPF2/UPF3 [93]. UPF1 contains an SQ (rich in serine and glutamine) domain at the C-terminus that blocks its helicase activity independent of the CH domain or phosphorylation [94]. Cancer-related mutations occur mainly in the CH domain and the ATP binding site [95]. Apart from its crucial role in nonsense-mediated mRNA decay (NMD), UPF1 is involved in DNA repair and replication [96,97]. Lower levels of UPF1 are seen in many cancer types, including pancreatic adenosquamous carcinoma [98], ovarian cancer [99], glioma [100], and HCC [101,102]. Lower UPF1 is associated with poor survival in lung adenocarcinoma [103], but its potential as a biomarker or therapeutic target requires further investigation.

### 2.2. Transcription Modifiers

*TP53* is the most frequently mutated gene in cancer, with over 50% of cancers carrying loss of function mutations [104,105,106]. The p53 tumor suppressor protein plays a central role in cellular stress responses and promotes cell cycle arrest, apoptosis, or senescence [104,105]. The N-terminal region contains two tandem transcription activation domains (TADs) required for target gene induction and tumor suppressor activities. The TADs are followed by a proline-rich domain that contributes to transcriptional activation and is essential for cell growth restriction [107]. Following this is the DNA-binding domain, which has been crystallized in complex with DNA and is the hotspot for cancer-associated mutations [108]. The oligomerization domain is essential for dimer/tetramer formation and together with the C-terminal domain facilitates, DNA binding [109]. It was shown many years ago that p53 can bind to its own mRNA at the 5′-UTR to regulate its expression [110]. p53 also binds the 5ʹ-UTR of *Mdmx* mRNA and the coding sequence of *Hspa5* (Bip) mRNA to regulate their translation [111,112]. The interaction of p53 with RNA may also regulate its oligomerization and DNA-binding activity [113].

IFN-γ-inducible protein-16 (IFI16) is a member of the PYHIN-200 family composed of one PYRIN domain and two HIN domains. The PYRIN domain serves as the location for homotypic protein–protein interactions. HIN domains are 200 amino acid motifs mediating interactions with DNA [114]. IFI16 acts as a DNA sensor involved in innate immune response to many viruses [115,116,117,118] and activates inflammasome formation upon infection [119]. IFI16 forms filamentous complexes on DNA and can recognize inverted repeats, DNA breaks, or G4 structures [120,121,122]. IFI16 is also involved in the response to RNA viruses such as Influenza A [123]. IFI16 may play a role in cancer by binding to p53, and thereby enhancing its transcriptional activity [124], and is involved in DNA damage responses via BRCA-1 interaction [125]. Higher levels of IFI16 are reported in renal cell carcinoma [126,127] and cervical cancer [128]. Its positive role was shown in HCC [129], head and neck squamous cell carcinoma [130], prostate cancer [131], and breast cancer [132]. As IFI16 is multifunctional, more studies will be needed to uncover its potential for diagnosis and treatment.

Topoisomerase II alpha (TOP2A) is an ATP-dependent enzyme that regulates the topological state of DNA during transcription. Binding to dsDNA is essential for stimulating its ATPase activity and the generation of dsDNA breaks. TOP2A is a key player in the decatenation checkpoint, and depletion leads to chromosomal mis-segregation. TOP2A is highly expressed in the G2 and M phases of the cell cycle [133] and is negatively regulated by p53 at the transcriptional level and by BRCA1 at the post-translational level via ubiquitination [133,134]. TOP2A was recently reported to associate with the transcription start site of numerous genes to regulate their transcription [135]. TOP2A is regarded as a marker of cell proliferation, and several of its interacting partners are associated with oncogenesis. Amplification or deletion of *TOP2A* and altered enzymatic activity occurs in numerous types of cancers [133,136,137,138,139].

Sex determining region Y-box 2 (SOX2) is a transcription factor belonging to the SRY homolog box 2 family and plays roles in pluripotency, somatic cell reprogramming, and neurogenesis. SOX2 complexes with other transcription factors, such as octamer-binding transcription factor 3/4 (Oct3/4), and binds to DNA motifs located adjacent to promoter/enhancer regions of several genes involved in development [140]. SOX2 can be post-translationally modified by methylation, acetylation, sumoylation, and phosphorylation, which control its function as a transcriptional regulator [141]. SOX2 also interacts with long non-coding RNAs (lncRNAs) involved in embryonic development and neural differentiation [142,143]. SOX2 can bind directly to ES2 lncRNA with high affinity through its DNA-binding HMG domain [144]. Numerous studies have reported aberrant expression of SOX2 in cancer (see review [145,146]), and *SOX2* is recognized as a powerful oncogene, where it regulates cancer stem cells [141].

AGO proteins are core components of the RNA-induced silencing complex (RISC), which is part of RNA interference (RNAi) processing. They bind guide RNAs (siRNA, miRNA) and process them for subsequent site-specific cleavage for the silencing of target mRNAs involved in development, differentiation, and protection against viral infection [147]. AGO2 consists of four domains: the N-terminal, PAZ, Mid, and PIWI domains. The PAZ domain binds single-stranded nucleic acids, the PIWI domain has an RNase-H-like fold denoting its endonuclease activity, and Mid shows similarity to the MC domain, which binds to the mRNA cap and is required for efficient translation [148]. The functions of AGO2 in tumorigenesis are diverse and range from high levels associated with poor prognosis in HCC [149], ovarian carcinoma [150], and gastric cancer [151], or mediating the elevation of oncogenic miR-378a-3p in Burkitt lymphoma [152], to its low expression as an indicator of poor prognosis in CRC patients [153]. AGO2 protein–protein interactions (e.g., KRAS and AGO2) are involved in the progression of pancreatic ductal adenocarcinoma [154] and NSCLC [155]. Studies of these multilevel effects are in progress and may offer patient specific targeted treatment, e.g., AGO2, as a delivery vehicle for inhibitory miRNAs [156].

Interleukin enhancer-binding factor 3 (ILF3) and ILF2 (also named NF45) were originally discovered as positive regulators of IL2 transcription as part of the NFAT-AP1-NF-kB enhanceosome in activated T-cells and have roles in cancer [157,158,159], autoimmune and inflammatory conditions [160,161], and psychiatric disorders [162]. ILF3 and ILF2 are involved in almost all steps of RNA metabolism, including transcription, post transcription, translation, pri-miRNA, lncRNA, circular RNA processing, and RNA editing [163,164]. ILF2 contains disordered N-terminal (amino acids 1–20) and C-terminal (amino acids 351–390) domains, and a central (amino acids 24–371) DZF (double strand RNA binding motif, dsRBM and zinc finger associated) domain. ILF2 forms heterodimers with ILF3, which itself contains two dsRBMs in the center and one tri-RG motif (three repeated RG sequences) [165] that interacts with ssRNA and ssDNA. In addition, two glycine-rich motifs (GQSY) in the C-terminus of ILF2 provide protein–protein interactions and are implicated in RNA granule assembly by association with mRNA ribonucleoprotein complexes [166]. ILF3/ILF2 dimers bind to chromatin and regulate expression of transcription factors that promote proliferation and suppress differentiation [167,168]. ILF3/ILF2 acts as a trans-regulator of RNA editing by interacting with adenosine deaminase RNA specific proteins (DAR1 and ADAR2, providing a link to cancer and epithelial-to-mesenchymal transition [169,170]). Further regulation of RNA metabolism involves stabilizing mRNAs by 3′UTR binding. ILF3 affects non-coding RNA activities and, for example, stabilizes miR-144 when bound to BUD13 or interacting with the lncRNA UBE2CP3/IGFBP7 duplex, which is linked to gastric cancers. ILF3 and ILF2 are overexpressed in several cancer types and contribute to tumorigenesis [171,172,173].

In addition to roles that are in common with ILF3, ILF2 also shows specific functions in tumorigenesis: In 1q21 amplified multiple myeloma, 1q21-driven ILF2 binds to YB1, interacts with the splicing factor U2AF65, and influences transcription of DNA damage response factors [174,175]. By interacting with E2F1, ILF2 promotes small cell lung cancer growth through maintaining mitochondrial quality [176]. ILF2 can bind to the regulatory region of the phosphatase and tensin homolog gene (PTEN) to promote anchorage-independence of NSCLC [177]. More recently, ILF2 was reported to bind the multiprotein transcription-export (TREX) complex component THOC4 and facilitate nuclear mRNA export via nicotine induced JAK2/STAT3 signaling in esophageal cancer [173].

Cell division cycle 5-like protein (CDC5L) was first discovered as a DNA-binding protein [178]. Its N-terminus contains a nuclear import site, DNA binding domains, and nuclear localization sites; the central region contains a hydrophilic proline-rich TAD, and the C-terminus preferentially associates with the spliceosome [179,180]. CDC5L and PRP19 (pre-mRNA processing factor 19) form a 700–1000-kDa complex with RBMX (RNA-binding motif protein X-linked), which is bound by lncRNA NORAD (non-coding RNA activated by DNA damage), ALYREF (Aly/REF export factor), and TOP1 (topoisomerase I). This complex plays important roles in genomic stability [181]. CDC5L is one of the core members of the pre-mRNA spliceosome [182] and is essential for cell cycle progression in yeast, plants, and mammals and in G2/M [183] and metaphase-to-anaphase transition in oocyte meiosis [184]. Phosphorylation by cyclin-dependent kinases is required for CDC5L-mediated pre-mRNA splicing [185]. Intriguingly, the CDC5L-Prp19/Pso4-Plrg1-Spf27 complex, which is involved in pre-mRNA splicing, also functions in DNA damage [186]. Prp19/Pso4 is modified by ubiquitination in response to DNA damage [187]. CDC5L acts as an adaptor, interacting directly with the cell-cycle checkpoint kinase ataxia-telangiectasia and Rad3-related protein (ATR) to recruit substrates to this E3 ligase complex [188]. CDC5L is highly expressed in glioma [189], HCC [190,191], and bladder cancer [192] and acts as an oncogene by regulating different target genes in different cancers. For example, in CRC, CDC5L binds to the promoter of telomerase reverse transcriptase (*TERT*) and regulates cell growth and proliferation through the PI3K/AKT pathway [193]. In melanoma, CDC5L upregulates fumarylacetoacetate hydrolase (FAH) transcription [194], binds the proline-rich receptor-like protein kinase (*PERK1*) gene promoter, and activates the ERK1/2 and JAK2 pathways in ovarian cancer [195]. CDC5L can also bind to RNA, for example, the lncRNA nuclear-enriched abundant transcript (NEAT1), which maintains prostate cancer cell growth [196].

### 2.3. Nuclear Organizers

CHTOP (chromatin target of protein arginine methyltransferase 1, PRMT1, also called small protein rich in arginine and glycine, SRAG, or Friend of Prmt1—Fop) was identified by two research groups as a nuclear/nucleolar protein [197,198]. The N-terminal region regulates localization within the nucleolus [197], and the arginine and glycine rich domain binds to PRMT1 [198]. CHTOP binds RNA and associates with facultative heterochromatin that affects estrogen responsive genes [197,198]. In addition, CHTOP is a component of the TREX complex and regulates alternative polyadenylation of target genes [199,200]. CHTOP is controlled through an autoregulatory negative feedback loop by an intron retention mechanism [201]. The characteristic chromatin binding of CHTOP hints at a role in epigenetic regulation of gene expression. In glioblastoma cells, the CHTOP-associated-methylosome complex binds to 5-hydroxymethylcytosine and methylates arginine 3 of Histone H4 (H4R3) to unwrap chromatin and transactivate cancer-related genes [202]. CHTOP is associated with apoptosis, stemness, and metastasis in chemoresistant epithelial ovarian cancer cells and may be a target to overcome chemoresistance [203,204].

HNRNPU is involved in many levels of gene regulation, including the maintenance of the chromosomal 3D structure [205,206], transcription, RNA splicing, and DNA repair. The N-terminal region (amino acids 1–160) is Asp/Glu rich and is the DNA binding site. HNRNPU also contains a nuclear localization site (NLS), a GX2GXGKT consensus sequence (amino acids 485–492) for putative NTP binding, an RNA polymerase II binding domain (amino acids 269–536), an actin binding site, and the RGG (Arg-Gly-Gly) region in the C-terminal region (amino acids 683–806) that binds to RNA [207]. These structural characteristics give rise to its diverse functions: the N-terminal acidic domain binds to matrix-associated region (MAR) DNA elements involved in chromosome organization [208]. By associating with different co-factors, it is involved in many regulation processes: HNRNPU can stabilize E3RS (an F-box protein, a receptor subunit of beta-TrCP ubiquitin E3 ligase) through the acidic N-terminal domain [209]. By cooperating with p300, HNRNPU binds to the scaffold/matrix-associated region (S/MAR) in the transiently silent topoisomerase I gene, where local acetylation of nucleosomes facilitates transcription [210].

The middle region of HNRNPU is bound by RNA polymerase II [211], colocalizes with Wilms’ tumor (WT1), and modulates WT1 transcription [212]. A short actin binding site close to the C-terminus of HNRNPU then further associates with the phosphorylated C-terminal domain of Pol II, thus carrying out actin’s regulatory role during the initial phases of transcription activation [211]. The RGG box is important in gene silencing, RNA splicing processing, and mRNA stabilization. HNRNPU co-localizes with the inactive X chromosome, which is RGG box-dependent [213,214]. As a nuclear organizer, it is not surprising that HNRNPU binds to almost all classes of regulatory noncoding RNAs, including all snRNAs required for splicing both major and minor classes of introns [215]. HNRNPU also plays important roles in DNA damage responses, in which DNA-PK phosphorylates HNRNPU at S59 to modulate protein–protein–RNA interactions that favor DNA repair enzymes [216]. One recent report showed that HNRNPU binds to telomeric G4 structures, thus regulating accessibility [217].

HNRNPU is involved in several cancers. In HCC, HNRNPU mediates the alternative splicing of Ras-related C3 botulinum toxin substrate 1 (Rac-1), yielding the variant Rac1b that stimulates tumorigenesis [218]. In CRC, hnRNPU and hnRNPA1 bind to the second exon of transformer 2-beta (TRA2β4) and upregulate the transformer 2 beta homolog (TRA2B) [219]. In neuroblastoma, HNRNPU activates the CCCTC-binding factor (CTCF) by binding to hepatocyte nuclear factor 4 alpha (HNF4A)-derived long noncoding RNA (HNF4A-AS1) [220]. The lncRNA, called HNRNPU processed transcript (also termed ncRNA00201), is up-regulated and provides oncogenic properties in pancreatic carcinoma [221].

Heterogeneous nuclear ribonucleoprotein L (HNRNPL) has a distinctive structure compared to other hnRNPs: The N-terminus is rich in glycine and contains two RNA-recognition motifs (RRM), RRM1 and RRM2; the central region contains a proline-rich linker, and the C-terminus contains RRM3 and RRM4. RRM1 has weak RNA binding. RRM3 and RRM4 are indispensable and are sufficient to bind to two appropriately separate binding sites within the same RNA by inducing RNA looping. RRM2 provides moderate RNA-binding affinity [222,223]. HNRNPL functions in DNA repair, RNA splicing, transcription, and translation, and is involved in several tumor types [224]. HNRNPL was found to directly regulate the alternative splicing of a set of prostate cancer-specific RNAs [225]. In lymph node-positive bladder cancer, HNRPL is recruited to the chemokine (C-C motif) ligand 2 (*CCL2*) promoter by lymph node metastasis associated transcript 1 (LNMAT1) and enhances transcription [226]. Similarly, lncRNA cancer susceptibility 9 (CASC9) forms a complex with HNRNPL that affects AKT signaling and DNA damage sensing in HCC [227]. LncRNA retinoblastoma associated transcript-1 (RBAT1) recruits HNPNPL to the promoter of the transcription factor E2F3 gene to upregulate its expression [228]. Most recently, HNRNPL was reported to facilitate the formation of circular Rho GTPase activating protein 35 (ARHGAP35) to promote cancer progression by interacting with the transcription factor TFII-I [229].

### 2.4. Signal Transmitters

PRKDC encodes the DNA-dependent protein kinase catalytic subunit, also called DNA-PKcs, which forms the catalytic sub-unit of the serine/threonine-protein kinase DNA-dependent protein kinase (DNA-PK). DNA-PK acts as a sensor of DNA damage, contributing to non-homologous end joining (NHEJ) and homologous recombination (HR) DNA repair pathways [230]. DNA-PKcs contains a large N-terminal helical domain, followed by the Circular Cradle that contains multiple HEAT (Huntingtin, Elongation Factor 3, PP2A, and TOR1) repeats, well-conserved phosphorylation clusters, and a C-terminal domain containing the highly conserved catalytic kinase domain [231]. DNA-PK was originally discovered as part of a transcription complex, but has been more extensively studied in DNA damage responses [230,232]. Aberrant expression and deregulated activity of DNA-PK is associated with numerous cancers and is correlated with poor prognosis [230,231].

UPF1, SMG1 (SMG1 nonsense-mediated mRNA decay associated PI3K related kinase) and SMG5 (SMG5 nonsense-mediated mRNA decay factor) are important parts of nonsense-mediated mRNA decay (NMD), a surveillance mechanism ensuring degradation of mRNA containing premature stop codons and regulating the stability of many wild type transcripts. In brief, NMD starts when there is an exon junction complex downstream of a stop codon. Eukaryotic peptide chain release factors 1 and 3, UPF1, and SMG1 are recruited to the stop codon site and form the SURF complex. Interaction of SURF with the exon junction complex leads to phosphorylation of UPF1 by SMG1 to activate its endonucleolytic activity or enable recruitment of SMG5 and SMG7 to mediate mRNA degradation [233,234]. NMD may have tumor suppressive or oncogenic activities [235]. Targeting NMD and thus enabling the translation of mRNAs with premature termination codons can lead to tumor suppression, as evidenced for CRC characterized by widespread instability in microsatellite sequences [236].

SMG1 is a member of the phosphoinositide 3-kinase related kinase (PIKK) family involved in NMD [237], DNA damage response [238,239], and telomere integrity maintenance [240]. Similar to its family member ataxia-telangiectasia mutated (ATM), it phosphorylates p53 on Ser15 upon DNA damage [238] and binds p53 mRNA under normal conditions, but dissociates upon ionizing radiation, leading to alternative splicing [239]. SMG1 is indispensable for alternative splicing of many mRNAs important during embryogenesis [241], and SMG1 haploinsufficiency plays roles in cancer development [242]. SMG1 can also be inactivated via promotor hypermethylation [243] or various miRNAs [244,245,246,247]. Low level SMG1 is associated with poor survival in HCC [248].

### 2.5. Telomere Organizers

TERT and telomerase RNA are the core components of telomerase, a reverse transcriptase serving to elongate chromosome ends. TERT is composed of TEN (telomerase essential N-terminal domain), TRBD (telomerase RNA-binding domain), RT (reverse transcriptase), and CTE (C-terminal extension domain) [249]. TERT activity is minimal in somatic cells, whereas ~90% of human tumors are characterized by telomerase activation [250] to provide cell immortality. Higher expression is caused mainly by *TERT* promoter mutations or focal amplification/rearrangements [251] and is associated with poor survival in many cancer types [252,253,254,255]. Therapeutic silencing of TERT activity is under evaluation, and several molecules have entered clinical trials [256].

SMG5 consists of an N-terminal 14-3-3-like domain important for heterodimerization with SMG7 [257], an α-helical domain, and a C-terminal PIN (PilT N-terminus) domain that is present in proteins with ribonuclease activity, but is inactive in SMG5 [258]. In addition to its role in NMD [259], as discussed above, SMG5 participates in telomere maintenance [240], is often upregulated in HCC, and is associated with poor prognosis [260].

## 3. Conclusions

It is not precisely known why some proteins are mutated only in certain types of cancer and what mechanisms determine this process. We show that DRBPs are differently altered across various cancer types. This can be explained either by different alteration loads in the type of cancer, or by the fact that DRBPs participate in cancer development in specific circumstances. We observed that some DRBPs show similar alteration tendencies across different cancers; for example, p53 and SOX2 are specifically affected across cancers. We searched the mutation sites across genes, but we did not reveal that the regions of DRBPs coding for DNA and/or RNA binding are preferentially mutated, thereby directly impairing their nucleic acid binding functions. In contrast, alterations occur randomly within these proteins, suggesting that the mutations influence overall protein structure more commonly than direct DNA/RNA binding activity, including the ability to interact with other partners in the functional DRBP complex.

This review has identified and assessed proteins that maintain DNA/RNA integrity through their dual binding properties and which are commonly altered in cancer. Some of these proteins are already therapeutically targeted, suggesting that they may all have potential for cancer treatment. In this context, it is important to realize that targeting a single protein will influence the entire spectrum of its interactions with proteins and/or nucleic acids. DRBPs are very diverse in their actions and inhibiting their activity could be beneficial for one process, but detrimental for another. Even for well-studied proteins, such as p53, the inhibition of its activity would suggest malfunction, but it can be also beneficial due to reducing the toxicity of radiotherapy and chemotherapy [261]. Therefore, successful introduction of novel agents that target DRBPs will require further identifying and understanding their interaction partners, signaling pathways, and mechanisms of regulation to develop multi-targeted personalized treatments for cancer patients. Obviously, the dual property of DRBPs provides a great spectra of tools for fine-tuning the processes in cells and therefore, more responsibility for alterations involved in cancer, regarding DNA and RNA binding proteins separately. We can study DRBPs consequences through combinations of different sophisticated techniques, e.g., selective 2′-hydroxyl acylation analyzed by primer extension (SHAPE), chromatin immunoprecipitation (CHIP), RNA immunoprecipitation (RIP), ribosome profiling (Ribo-Seq), and next generation sequencing (NGS). The future research should be more focused on interconnections between DRBPs in cancer.

## Figures and Tables

**Figure 1 ijms-23-09329-f001:**
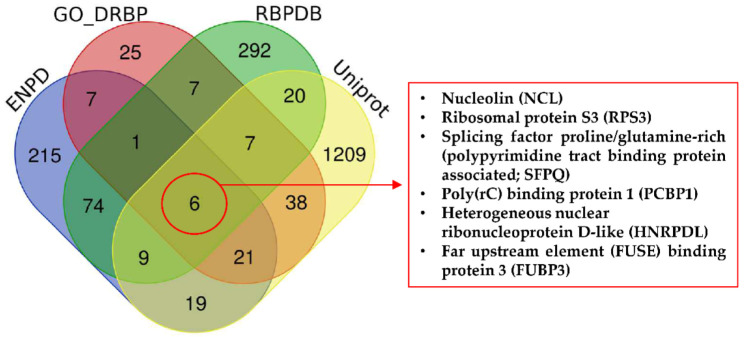
Cross-search analysis of DNA/RNA binding proteins in selected databases. RBPDB, the database of RNA-binding protein specificities, contains 1171 proteins (424 proteins from humans). The database is focused on RNA-binding proteins and was last updated 21 November 2012 v1.3.1. ENPD, the eukaryotic nucleic acid binding protein database, is a library of nucleic acid binding proteins, and their functional information considers DNA binding proteins, RNA binding proteins, and DNA and RNA binding proteins. The database includes 2736 DNA and RNA binding proteins. The last update was released 1 October 2018 (v1.01). GO, the gene ontology knowledgebase, contains information on gene functions. The last release date was 1 June 2020. Uniprot considers 562,755 manually annotated and reviewed proteins (last release February 2020).

**Figure 2 ijms-23-09329-f002:**
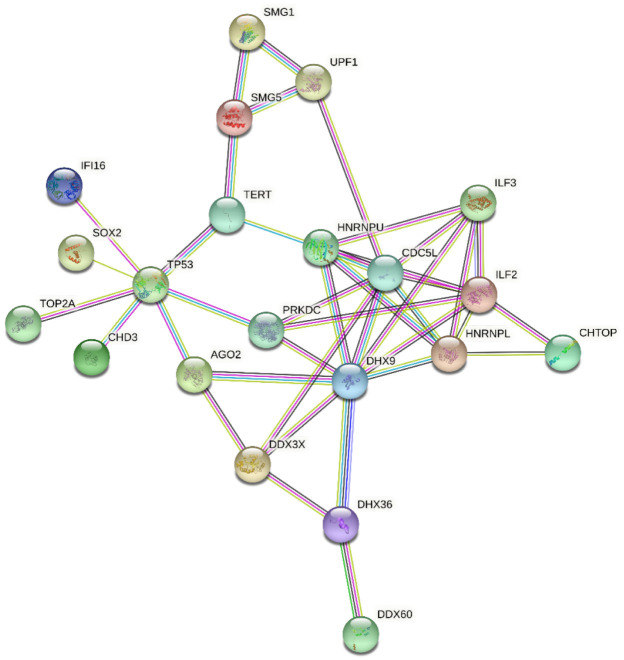
Interaction net of the most altered DNA/RNA binding proteins involved in cancer.

**Figure 3 ijms-23-09329-f003:**
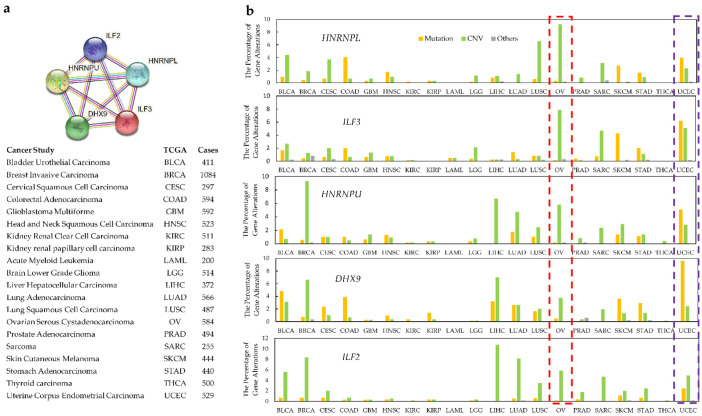
Specific pattern of alterations in a narrower cluster within the interaction network. DRBPs within a close association network show cancer type-specific alteration. Cancer types from TCGA (n ≥ 200 samples per type) were used for DRBP alteration frequency analysis. (**a**) The functional association network of HNRNPL, ILF3, HNRNPU, DHX9, and ILF2. (**b**) Alteration categories for these genes in specific cancer types. Ovarian serous cystadenocarcinoma is highlighted in the red boxed region and uterine corpus endometrial carcinoma in the purple boxed region.

**Figure 4 ijms-23-09329-f004:**
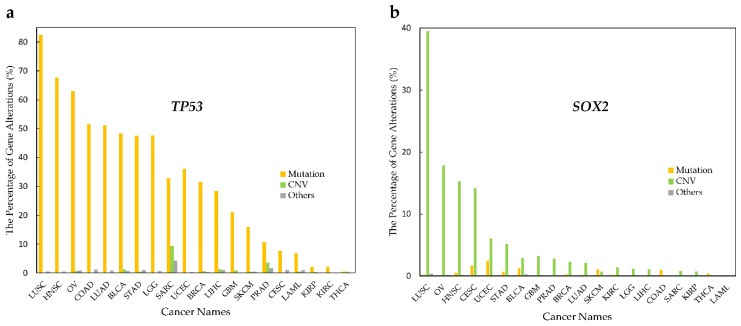
Alteration type in p53 and SOX2 in various cancer types. Alteration frequency in p53 and SOX2 by cancer type. (**a**) *TP53* mutations rates are higher than CNV rates in all cancer types. (**b**) *SOX2* shows higher CNV rates than mutation rates in most cancer types.

**Figure 5 ijms-23-09329-f005:**
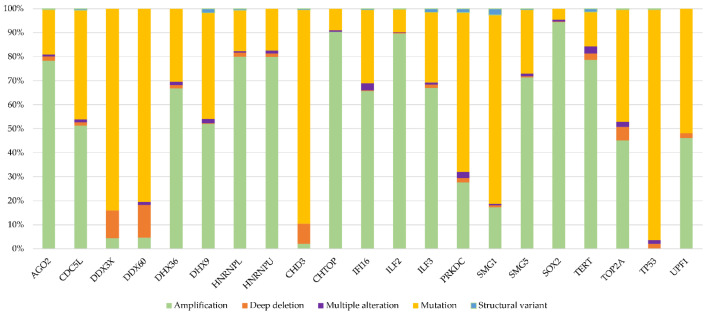
Distribution of alteration types in selected genes; the percentage of alteration types from the total number of alterations in the given gene (TCGA).

## Data Availability

Not applicable.

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
