# Peer review of "DNA and RNA Binding Proteins: From Motifs to Roles in Cancer"

_ijms, 2022, doi:10.3390/ijms23169329_

Round 1

Reviewer 1 Report

I was very pleased to see the submitted review that provides an overview of DRBPs highlighting their characteristics and their involvement in diseases like cancer. The authors used various protein databases to identify the different DRBPs involved in cancer. The description of the DNA, RNA and DRBP motifs and structures are well described and elaborated. All the classifications of the motifs and protein domains are included in this review. It is quite impressive how the authors used different bioinformatics tool to narrow down and find the six different DRBPs involved in cancer. The authors also highlighted the involvement of mutations, copy number, transcription factors and variation amongst different cancers to correlate the association between cancers and DRBPs. Towards the end of the review, the authors also described extensively a list of 21 DRBPs according to their function and assessed their characteristics. In addition, these DRBPs were also studied to demonstrate as a possible potential targets for therapeutic intervention. This review will definitely help in understanding more about DRBPs and their implications in cancer. The review is straightforward, well-written and conveys the message clearly. Here are my comments.

1.       The structural features of DRBPs have remained remarkably stable throughout evolution, indicating that dual nucleic acid binding confers selective advantages. The evolution of nucleic acid sequences plays an important role in the development of DRBP function. Emerging research on rapidly evolving lncRNAs suggests that RNA binding by previously thought to be DNA-specific proteins may be a widespread phenomenon. When DNA and RNA compete for the same DRBP interaction surface, DRBPs may have less stringent criteria for interacting with nucleic acids or may adjust their structure. Please include a sentence or two on whether future studies will be able to find more involvement of DRBP in regulating the behavior of cells in cancer. Do you agree if more sophisticated technologies will  be required to identify more DRBPs along with the existing technologies and what improvements may be required?

2.       DRBPs make up a sizable portion of cellular proteins and play critical roles in cells. Their functions include transcription and translation control, DNA repair, splicing, apoptosis, and stress response mediation. Binding to both DNA and RNA allows DRBPs to integrate multiple signals into cellular signaling networks, improving gene targeting, fine-tuning gene expression, and incorporating metabolic states or stresses to modulate protein activity. Please include a sentence or two whether DRBPs are majorly responsible for alterations involved in cancer over DNA and RNA binding proteins separately.

3.       Please arrange the keywords alphabetically.

Reviewer 2 Report

The review article by Bonczek et al. on the DNA and RNA binding proteins focuses on the critical role of those proteins in cancer progression. The article is well written with self-explanatory figures. However, I have few suggestions to make. 

1. In the section 2 authors performed searches in four different databases to report the DNA/RNA binding proteins that are shared in all the four databases. The cancer is one of the well-studied disease and there are several annotated repositories for the cancer proteins. In fact, there are repositories that are specific to cancer types. Digging those data might end up in increased number of DNA and RNA binding proteins. 

2. In addition to figures, I would recommend authors to include tables. It would be nice to summarize all the proteins, functions and their prevalence in cancer types in a single table which would help in easy readability.

3. Regarding the subsections in section 2, some are named after the protein types example helicases, kinases; some others are named after the protein functions example transcription modifiers. I would suggest authors categorizing proteins based on the single criteria i.e. either categorize based on function or type not both.

Reviewer 3 Report

In this review article, the authors employed bioinformatics to identify sets of DNA/RNA binding proteins, and their genetic alteration might be necessary for cancer development. However, the description is too general and fragmented. The authors might need to consider if there is any common thing that can connect these candidates rather than they can interact with nucleic acids.

Specific points:    

1.       Regarding figure 5, does it result from a particular type of cancer or a combination? Also, based on the figure, does it mean amplification of AGO2 appears in 80% of cancer? Or do just 80% of the cases with the alteration belong to amplification?

2.        In figure 3, the authors analysed the type of genetic alteration in 4 examples. How about the other genes?

3.       In section 2, the authors described the nature of each candidate and mentioned their roles in different cancers. Do these candidates share common features, or are they engaged in particular pathways to promote cancer development or progression e.g. local recurrence and metastasis or drug resistance? The authors might need to reorganise section 2 to give more meaningful insight rather than just stating the facts.  

4.       I would suggest the author focus on a particular candidate with DNA/RNA binding domain and discuss the roles of the DNA binding protein in cancers. Moreover, identify if inhibiting DNA binding or nuclear import would suppress cancer development.    

Round 2

Reviewer 3 Report

The authors have already addressed all my concerns.